



# High-resolution soil moisture mapping in northern boreal forests using SMAP data and downscaling techniques

Emmihenna Jääskeläinen[1], Miska Luoto[2], Pauli Putkiranta[3], Mika Aurela[1], and Tarmo Virtanen[3]

[1]Finnish Meteorological Institute, Erik Palmenin aukio 1, 00560 Helsinki, Finland
[2]Deparment of Geosciences and Geography, University of Helsinki, P.O. Box 64, Helsinki 00014, Finland
[3]Ecosystems and Environment Research Programme, Faculty of Biological and Environmental Sciences, University of Helsinki, P.O. Box 65, Helsinki 00014, Finland

**Correspondence:** Emmihenna Jääskeläinen (emmihenna.jaaskelainen@fmi.fi)

**Abstract.**

Soil moisture plays an important part in predicting different forest-related phenomena, such as tree growth or forest fire risk. As these phenomena influence the carbon storage capacity of boreal forest ecosystems, it is crucial to provide soil moisture information at high temporal and spatial scales. Current satellite-based soil moisture products often have high temporal resolu-
5    tion at the expense of spatial resolution. Therefore, we developed a machine-learning-based model to estimate soil moisture at high temporal and high spatial resolution over boreal forested areas for the annual time period from May to October. The basis data of the model is the enhanced 9 km spatial resolution soil moisture data from the Soil Moisture Active Passive (SMAP) mission. Additionally, soil and vegetation properties, reanalysis-based parameters, and measured in situ soil moisture data are used to guide the model construction process. The analysis of the developed model shows that the model retains the temporal and large-scale spatial variability of SMAP soil moisture. Furthermore, comparisons with the independent in situ soil moisture
10    data show that the soil moisture values predicted by the developed model have a better agreement with in situ values than SMAP soil moisture, as RMSE decreases from 0.097 $m^3/m^3$ to 0.065 $m^3/m^3$, and correlation increases from 0.30 to 0.52 over forest sites. Therefore, this machine-learning-based model can be used to predict high-resolution soil moisture over boreal forested areas.

## 1  Introduction

Boreal forest ecosystems are important carbon sinks and stocks (Pan et al. (2011), Pan et al. (2024)). Trees, mineral soil, and organic layer account for about 70% of the carbon pool in boreal forests (Merilä et al. (2023)). Trees remove carbon dioxide from the atmosphere through photosynthesis, turn it into organic carbon compounds, and use them for growing. Carbon is stored in all parts of the tree, i.e. in branches, stems, leaves, bark, and roots (e.g. Clemmensen et al. (2013), Thurner et al. (2014)).
20    This carbon stored in the boreal ecosystems is released back into the atmosphere for example due to forest fires (Walker et al. (2019)) and the decomposition of trees, turning forests from carbon sinks to sources. As soil moisture plays a significant role in predicting tree growth (Larson et al. (2024)), forest fire risks (Walker et al. (2019)), and carbon stock partitioning (Larson et al. (2023)), it is essential to provide soil moisture data at a large spatial scale and high temporal frequency across boreal forests.



Due to the considerable local variation in soil moisture and the sparsity of the in-situ measurement network, the only
viable way to extensively observe soil moisture over boreal forests is to use satellite-based soil moisture data sets. However,
persistent cloud cover, other weather-related phenomena, and high solar zenith angles hinder the use of optical satellite-based
soil moisture data, making microwave-based soil moisture the most feasible option. For example, Sentinel-1 C-band Synthetic
Aperture data (SAR) has been used to retrieve soil moisture with high resolution (e.g. Bauer-Marschallinger et al. (2019),
Balenzano et al. (2021), Manninen et al. (2022)), but the dense vegetation prevents radar signal from reaching the soil surface
(Flores et al. (2019)), and thus causes uncertainty in the results (Bauer-Marschallinger et al. (2019), Flores et al. (2019)). A
longer wavelength band, like L-band, can penetrate the vegetation all the way to the soil surface (Flores et al. (2019)), and
possibly also deeper than the documented -5 cm depth in the boreal forest (Ambadan et al. (2022)). The well-known L-band-
based soil moisture mission Soil Moisture Active Passive (SMAP, https://smap.jpl.nasa.gov/) has been measuring soil moisture
globally from 2015 onwards, and has been reported to be sensitive to soil moisture changes under the forest canopy (Colliander
et al. (2020), Ayres et al. (2021)). The disadvantage of soil moisture data from SMAP is that the original spatial resolution is
very coarse, with grid size being 36 km (Entekhabi et al. (2014)). SMAP soil moisture data has been enhanced to 9 km, but as
soil moisture can change a lot even in a short distance (Mälicke et al. (2020)), there is a need for soil moisture data in a finer
spatial resolution.

Instead of directly using satellite-based measurements to retrieve soil moisture, another approach is downscaling. This
involves enhancing coarse-resolution soil moisture data to a finer spatial scale using regression or more advanced machine
learning methods. Downscaling has been used widely (e.g. Peng et al. (2017), Sabaghy et al. (2018), and references therein)
with promising results. A few examples of downscaled soil moisture in 1 km spatial resolution include GLASS SM (Zhang
et al. (2023)), which is based on ERA5-Land soil moisture; an over 20-year gap-free global and daily soil moisture data set
(Zheng et al. (2023)) based on ESA-CCI soil moisture; and downscaled SMAP (Fang et al. (2022)). Since the original data
sets have very coarse spatial resolutions, the downscaled data sets typically aim for a spatial resolution of 1 km or coarser.
However, a disadvantage of these downscaled data sets is that they are not specifically designed for boreal forests.

In this study, we provide a model to calculate a high resolution (1 km and 250 m pixel-sized) soil moisture for boreal forests.
We use SMAP soil moisture data in 9 km spatial resolution as the basis, and we combine SMAP soil moisture with in situ
soil moisture observations. Other high-resolution and soil moisture linked parameters (like vegetation and soil properties) are
added to our downscaling machine learning model to guide the process. The coarse resolution of SMAP, the small amount of
in situ measuring sites over boreal forest (i.e. low variability in static input parameters), and the mainly forested locations of
the in situ sites, hinder the model construction process and therefore our model is limited to forested areas, excluding other
areas like peatlands.

This paper is constructed as follows. First, all the used data sets are introduced in Section 2, followed by preprocessing
steps and model construction in Section 3. In Section 4, the results of the model analysis and model validation are shown. We
conclude with a discussion and conclusions in Sections 5 and 6, respectively.



## 2 Data

In this study, we used in situ data from two larger boreal forests in situ observation areas with easily accessible data, one in Northern Finland (NF), operated by the Finnish Meteorological Institute (FMI), and the other in Alaska, operated by multiple
networks (see section 2.7 below). For model construction, we decided to use NF sites, leaving the Alaska sites for validation. The locations of the sites are shown in Figure 1. Based on the observations between the years 2019 and 2023 from the weather stations located in the NF (https://www.ilmatieteenlaitos.fi/havaintojen-lataus), there is snow cover typically from mid-October to May, depending on the site and location. Therefore, for the study time period, we chose the annual period spanning from the first of May to 15 of October, covering the years 2019–2023. From here onwards, the use of the soil moisture term indicates
volumetric water content (%).

### 2.1  SMAP soil moisture

The SMAP mission was meant to combine radiometer (passive) and radar (active) observations. However, since the radar broke down just months after the launch, the radiometer is currently the only instrument observing the surface. This SMAP L-band (1.41 GHz) radiometer has a native spatial footprint of 36 km, but the footprint is enhanced to 9 km resolution by using
the Backus-Gilbert optimal interpolation algorithm (Chan et al. (2018)). The enhanced products are provided on the global cylindrical EASE-Grid 2.0 (Brodzik et al. (2012)).

SMAP soil moisture is based on retrieved brightness temperature data in horizontal and vertical polarizations (O'Neill et al. (2021)). Water body correction is applied to the brightness temperature data first to remove water bodies, as they lower the brightness temperature values and hence cause overestimated soil moisture values. Then tau-omega-model (tau, vegetation
optical depth $\tau$ and, omega, vegetation single scattering albedo, $\omega$) is applied to the single channel (horizontal and vertical) brightness temperature data to separate soil and vegetation contributions from the total brightness temperature. After that, soil moisture is retrieved by inversion from the tau-omega corrected brightness temperature. Land use classification data is used to determine the $\tau$ and $\omega$ values for different areas. For dual-channel retrieved soil moisture, the tau-omega corrected single-channel brightness temperature data is used.
In this study, we use the enhanced SMAP L3_SM_P_E product (O'Neill et al. (2023)), in which the global surface soil moisture (0–5 cm) in $m^3/m^3$ is provided twice a day, at 6:00 am (descending) and at 6:00 pm (ascending). Three different soil moisture products are available, one calculated from each single channel and one dual-channel product. As the latter one is currently the baseline product (Chan and Dunbar (2021)), we chose that for this study. Further, we chose the north polar maps as they cover the boreal zone, and we focus on the soil moisture data at 6 am (descending overpasses). This SMAP data in
9 km resolution is used as an input for the soil moisture model.

### 2.2  MODIS

The Moderate Resolution Imaging Spectroradiometer (MODIS) instruments are aboard the Terra and Aqua satellites, which were launched in 1999 and 2002, respectively. As Sun-synchronous satellites, they provide almost global coverage every 1 to 2





days. Terra is set to a descending orbit (measurements at 10.30 am) and Aqua to an ascending orbit (measurements at 1.30 pm).

The MODIS instrument measures multiple wavelength bands, resulting in a wide range of obtained parameters. In this study, we use vegetation indices from both MODIS instruments. We use MYD13Q1 (from Aqua, Didan (2021a)) and MOD13Q1 (from Terra, Didan (2021b)) products (version 6.1) which are global 16-day-mean data sets with 250 m spatial resolution. The used data, enhanced vegetation index (EVI) and Normalized Vegetation Index (NDVI), are provided in the Sinusoidal tile grid. EVI and NDVI contribute to vegetation effects of the soil moisture model.

## 2.3 SMAP-based 1 km soil moisture data

SMAP, enhanced to 9 km spatial resolution, was further downscaled to 1 km (Lakshmi and Fang (2023)) by using thermal inertia theory (Fang et al. (2022)). Based on that theory, the land surface temperature (LST) difference between night and day is negatively correlated to the soil moisture. For downscaling SMAP, the MODIS LST data in 1 km spatial resolution from Terra (night) and Aqua (day) were used, combined with the NDVI, also from MODIS. The NDVI, divided into 10 groups by

using an interval of 0.1, is used for grouping soil moisture and LST differences. The assumption behind this is that changes in NDVI affect the relationship between soil moisture and LST difference. Based on the validation, the downscaled SMAP data performs better in low latitudes and warm months, compared to high latitudes and cold months (Fang et al. (2022)). The SMAP in 1 km resolution is used in this study as an example of downscaled data based on SMAP in 9 km resolution.

## 2.4 ERA5-Land

ERA5-Land, the land component of the fifth generation of European Centre for Medium-Range Weather Forecasts (ECMWF) atmospheric reanalysis of the global climate (ERA5), is produced by the Copernicus Climate Change Service. Similarly to ERA5, the land component also covers the period from 1940 to the present day (Hersbach et al. (2020)), but with enhanced spatial resolution (from 31 km to 9 km). ERA5-Land provides hourly data of various surface parameters, of which we used total precipitation (m), air temperature at 2 m above the surface (K), and surface net solar radiation ($J/m^2$) as inputs for the soil

moisture model. They provided the weather contribution to the model.

## 2.5 Soil properties

SoilGrids version 2.0 provides multiple soil properties globally in 250 m spatial resolution. These data sets are constructed by using quantile regression forests to about 240 000 in situ soil observations and numerous environmental covariates (Poggio et al. (2021)). From the resulting data sets, we chose bulk density ($cg/cm^3$) and silt content (g/kg) at the depth interval 0–5

cm to be used as inputs in models. As static inputs, they provide the invariable soil contribution to the model. SoilGrids also provide uncertainty information for each soil property, and therefore they were acquired as well.





## 2.6 CORINE land cover

The Coordination of Information on the Environment (CORINE) program was launched in the 1980s, as there was a need for detailed and harmonized land cover data set over the European continent (Büttner et al. (2017)). The current land cover data

covers the pan-European area with 100 m spatial resolution. The data set consists of 44 classes, and it is updated every six years. In this study, we use CORINE land cover data from 2018 to determine the land use classifications of the study area, the land cover classes of the used in situ sites in the NF area, and we also used land cover data to create a mask to exclude water bodies and all the other land covers except forested areas.

## 2.7 In situ data

In situ soil moisture data for model training and testing are from the Arctic Space Centre of the Finnish Meteorological Institute (FMI-ARC, https://fmiarc.fmi.fi/). FMI-ARC hosts a measurement infrastructure, which is used to monitor, for example, the atmosphere, soil properties, snow properties, precipitation, and carbon and water cycles. All collected observations can be found at https://litdb.fmi.fi/ (last accessed: 30 September 2024). For in situ soil moisture observations, the measurement sites are located around Sodankylä and Saariselkä, and they cover mostly boreal forested sites. The chosen in situ sites with additional

information can be found in Table A1, and their locations are shown in Figure 1. The in situ soil moisture is measured at different depths, and for this study, we chose a depth of -5 cm.

The in situ data for validation of the constructed model are located in the boreal zone in Alaska. In situ soil moisture data has been collected in the International Soil Moisture Network (ISMN) database starting from 1952 (Dorigo et al. (2013), Dorigo et al. (2021)). In situ soil moisture data is provided to the ISMN by multiple organizations for free use. We chose 16 stations

located in the boreal zone (Figure 1, right side), and information about those sites can be found in Table A2. We focused on the year 2019, as all 16 stations had observations during that year. Similarly to NF sites, the in situ soil moisture is also measured in different depths, and for this study, we chose a depth of -5 cm.

## 3 Methods

### 3.1 Preprocessing

All gridded data used (SMAP, EVI, NDVI, ERA5-Land data, and soil properties) are reprojected to EASE2-grid if needed and resampled to achieve a spatial resolution of 1 km. This means that the projection matches that of SMAP, but the spatial resolution is finer than that of SMAP. If the original resolution is coarser than the resampled one, the resampling is done by using the nearest neighbor. On the other hand, if the original resolution is finer than the resampled one, then the resampling is done by taking the average of all values within the coarser pixel. The average is taken even if there is only one value within the

coarser pixel. This ensures that the model inputs have a minimal number of missing values.

After resampling and reprojecting, some of the data are further preprocessed. As EVI and NDVI data from both MODIS instruments are originally provided every 16 days, we obtain daily maps of EVI and NDVI by linear interpolation over time





using the closest available observations. The linear interpolation was chosen because it is easy to implement and does not cause any major discrepancies in the interpolated data for vegetation types with weak seasonal changes, such as evergreen needle-leaved forests (Li et al. (2021)). After interpolation, we calculate the mean value of Terra and Aqua -based vegetation maps to obtain only one EVI and NDVI map per day. Precipitation, air temperature, and surface net solar radiation from ERA5-Land are provided for every hour. Instead of using instantaneous precipitation and air temperature reanalysis values, we calculated a sum over 72 hours before each SMAP observation. This approach takes into account the cumulative effects of temperature and precipitation. From surface net solar radiation, we used instantaneous values only. In situ data for training and testing was cleaned by removing those stations and those years where soil moisture values were abnormally low (below 0.05 continuously, or decreased to zero regularly), as including those values might lead to the model underestimating soil moisture.

After preprocessing and data cleaning, all the gridded data are matched with NF in situ locations. If there are multiple in situ values within the same 1 km pixel, we take a mean value of those soil moisture values and use that instead to represent the soil moisture in that location. By doing this, we end up with only 14 individual locations, as most of the in situ sites are located near each other. During the individual location acquiring process, one of the two peatland sites (SUO0004) ended up being the only site in one 1 km pixel with extremely high in situ soil moisture values ($> 0.8$ m³/m³). Including that location in the training set caused the model to predict erroneous soil moisture values. Therefore, that site was excluded from the study data set.

### 3.2 Model for soil moisture

The data set for model construction consists of only 13 individual locations, meaning that there are only 13 different bulk density or silt content values. We aimed for high variability in soil properties in the training set and, conversely, low variability in the test set to produce a model that is as generalized as possible using the available data. Therefore, we chose 9 of those 13 sites for the training data set based on their soil properties. The other 4 were left for the test set. In the training set, the bulk density values vary from 0.57 cg/cm³ to 0.66 cg/cm³, and silt content from 27.7 g/kg to 34.6 g/kg. In the test set, the bulk density varies from 0.55 cg/cm³ to 0.63 cg/cm³, and silt content from 27.8 g/kg to 30.5 g/kg. The placing of the individual in situ sites to training or test set is shown in Figure 1 and Table A1.

We used all the available data from the chosen annual periods covering the years 2019–2023, and hence we had 4480 values for training and 2221 for testing. Tree-based algorithms are commonly used in soil moisture predictions (e.g.Wei et al. (2019), Tramblay and Quintana Seguí (2022), Ning et al. (2023), Shokati et al. (2024)), and it has been reported that tree-based methods can outperform deep-learning methods (Li and Yan (2024)). The Gradient Boosting (GB) method (Breiman (1997), Friedman (2001), Friedman (2002)), in which the weak learners (decision trees) are trained sequentially by correcting the residuals of the previous model, was therefore chosen for model construction. We used a framework for tree-based algorithms called Light Gradient-Boosting Machine (lightGBM), as it is faster to use (Ke et al. (2017)).

We hypertuned the model parameters by using the GridSearchCV method from scikit-learn (Pedregosa et al. (2011)). It is a method where all possible combinations of given model parameters and their grids are tested and evaluated by using cross-validation. In our model building, we used CV=3. The chosen parameters with their test ranges are shown in Table 1. The



learning rate was chosen to be 0.05, and we set the number of trees to 200. We also used a feature fraction of 0.4 to limit the overfitting and prevent one static input (bulk density or silt content) from having too much importance over that of the other inputs.

## 4 Results

### 4.1 Analysis of the model

The SHapley Additive exPlanations (SHAP, Lundberg et al. (2020)) values (which specify the effect of different individual inputs on the output) indicate that static inputs dominate the results, as can be seen from Figure 2. Bulk density and silt content have a higher influence on the predicted soil moisture compared to other inputs. Also, their effect is nonlinear. On the other hand, SMAP soil moisture has a clear linear effect on the results, although smaller (compared to static inputs). The rest of the inputs (EVI, NDVI, and ERA5-Land based) have much lower impacts but are significant enough to be included as inputs.

The RMSE, R, and $R^2$ values between the training and test set indicate at least small overfitting when using all test data (Table 2). Due to the small size of the training and test sets, along with low variability in static inputs, preventing overfitting is challenging. These limitations also make the statistical values sensitive to changes in the data sets. One site (DIS0004 spots 1 and 2, in Test site D) had a clear deviation from the SMAP values in the test data set, and if we remove those values from the test set, overfitting decreases remarkably, which can be seen in Figure 3.

As the original highest spatial resolution of some inputs is 250 m (NDVI, EVI, and soil properties), we also resampled SMAP soil moisture and ERA5-Land based inputs to that same 250 m spatial resolution using nearest neighbor resampling. We then calculated soil moisture maps from those 250 m resolution data maps using the constructed GB model to study how sensitive the developed model is to small changes in vegetation and soil property values (i.e. as those are the only parameter values changing within one time step). Exemplary time series for NF test sites for the year 2023 are shown in Figure 4. The individual in situ sites are located close to each other and therefore the Test sites A-C have the same in situ sites in both resolutions. Only in Test site D one site (DIS0004 spot2) locates in a different pixel. Due to that, in situ soil moisture values for 250 m resolution in site D would be systematically about 0.04 m³/m³ lower compared to the in situ soil moisture values in 1 km resolution. Because of only a small difference between in situ values in different resolutions, we omitted the in situ soil moisture in 250 m resolution from the test site D in Figure 4. Overall, as all sites (A-D) are boreal forest sites, SMAP soil moisture is well in line with in situ soil moisture values. Furthermore, predicted values calculated for both 1 km and 250 m resolution data are in line with in situ values and SMAP soil moisture, even though predicted data in 250 m resolution can have systematic differences (Test site B). Based on these results for NF sites, the developed model is not overly sensitive to small changes in soil and vegetation properties data. Also, based on these time series results, the developed model detects temporal changes well. In hindsight, as the model is constructed using SMAP soil moisture, and SMAP soil moisture data is noisy, the same noisy features can be found in predicted values. Also, due to the SMAP being the basis for the developed model, the predicted values have the same temporal resolution as SMAP, meaning that data can be predicted almost daily if SMAP soil moisture data are available.



We also calculated the soil moisture values for the whole NF area using the constructed model to analyze how well the model captures the spatial variations and also to show the impact of missing pixels on the predicted maps. We calculated soil moisture maps using 1 km and 250 m resolution data. Examples of these predicted soil moisture maps are shown in Figures 5 and 6. Predicted soil moisture values are lower than SMAP soil moisture values, and for 250 m resolution maps the number of missing pixels increases. Nevertheless, spatial changes are well detected by the predicted values when compared to SMAP soil moisture. The missing values in predicted maps are due to the missing data in inputs. SMAP data have missing data because of water bodies or otherwise failed soil moisture retrievals, whereas soil property data are missing only due to water bodies. Similarly, vegetation properties are not retrieved over water bodies, but vegetation data are also missing because of missing measurements, caused typically by cloud cover (as vegetation properties are based on optical data). Furthermore, as the model is developed mainly for forested areas, a land cover mask was applied to the results (shown only in Figure 6, and omitted in Figure 5 for clarity). We used CORINE land cover data in 100 m spatial resolution as the basis of the mask. Land cover data was resampled to the 1 km and 250 m spatial resolutions and those pixels where forest classes covered under 50% of the coarser pixel were masked.

## 4.2 Model uncertainty

We used the sensitivity of the most important inputs and the standard deviation of the difference between predicted soil moisture values and in situ values from test data as the uncertainty of the model. First, we approximated the uncertainty each important input causes to the results. Predicted soil moisture from the training data was used as the reference data. Then we added errors to the important inputs separately from their error distributions $\epsilon \sim \mathcal{N}(0, \sigma^2)$. For soil properties, the $\sigma$ was taken from their uncertainty estimations. For vegetation indexes, we used the reported uncertainties, 0.015 for EVI and 0.025 for NDVI (https://modis-land.gsfc.nasa.gov/ValStatus.php?ProductID=MOD13). For SMAP soil moisture, we used the standard deviation from the difference between SMAP soil moisture and in situ soil moisture from the whole in situ data set from NF. The obtained standard deviation was 0.12. For ERA5-Land based inputs, we decided to omit their uncertainty, as their impact on the model is small. We calculated the difference between the error-added values and the reference data 100 times. The sensitivity of each varied input, the test std, and the total uncertainty for the constructed model are shown in Table 3. The total uncertainty is calculated as a squared sum between the individual sensitives and test std, that is:

$$\epsilon = \sum_{i}^{N} (u_i^2). \tag{1}$$

The sensitivities of each input are in line with their importance to the model. Bulk density and silt content have the highest impact on the model uncertainty for individual inputs. On the other hand, vegetation properties have the lowest impact. In total, the model uncertainty is around 0.119 $\mathrm{m}^3/\mathrm{m}^3$.





## 4.3 Validation with Alaska sites

Altogether 16 stations from Alaska were used as an independent model validation set. We calculated statistical values (RMSE, URMSE, and R) for each site between in situ soil moisture and SMAP in 9 km resolution, SMAP enhanced to 1 km resolution, and GB-model-based predicted values, both 1 km and 250 m. SMAP in 1 km resolution has a slightly lower mean RMSE compared to others, but predicted values in 250 m have the lowest mean URMSE value (Figure 7). GB-model-based predicted values in both 1 km and 250 m resolution have clearly higher R values compared to SMAP soil moisture data sets.

Of the 16 stations, only 4 were reported to be located in forested sites (information is based on ESA CCI Land Cover (ESA (2017))). Soil moisture data comparisons from those four sites are shown in Figure 8. SMAP soil moisture in both resolutions has a lot of variability compared to predicted estimates. Predicted values are closer to the in situ values.

The time series for summer 2019 (May to October) for sites Aniak and Gulkana River (tree-covered sites) are shown in Figure 9. For Aniak (left side), SMAP in 9 km resolution and predicted values in 1 km and 250 m resolution are most of the

time well in line with in situ values. SMAP in 1 km has lower soil moisture values compared to the in situ values. For the Gulkana River site (right side), all data sets have difficulties at the beginning of the summer. The high soil moisture values are probably due to the snow melting increasing the soil moisture, which is not detected by SMAP. Predicted values catch the high values at the beginning of the summer slightly better than SMAP soil moisture data sets. Predicted values in 1 km have too much temporal variability, which is missing from the 250 m data.

Sixteen in situ sites in Alaska were grouped into coarser land use classification classes (forest, mosaic, shrub, and sparse), and RMSE, URMSE, and R values were calculated between in situ values and each satellite-based data, the values are shown in Table 4. For forested sites, predicted values in 250 m have the lowest RMSE and URMSE, and highest R values compared to other data sets. Predicted values in 1 km resolution have the second-highest model validation statistics. For mosaic sites, SMAP in 9 km has the lowest RMSE, but the predicted values have the highest URMSE and correlation values. All data

sets struggle to predict soil moisture values in shrub and sparse sites. Based on these validation results, the developed model predicts temporal changes relatively well.

## 5 Discussion

Spatio-temporal data on the variation in soil moisture for boreal regions is crucial for predicting forest-related phenomena, such as tree growth and forest fire risk, both of which influence the carbon storage capacity of these ecosystems. However,

existing satellite-based soil moisture products for vegetated areas often have coarse spatial resolution. To address this issue, higher-resolution data is necessary to capture the finer spatial variations in soil moisture. Consequently, we developed a model utilizing satellite data to estimate soil moisture at high resolution (1 km and 250 m) over boreal forested regions. We used a tree-based machine learning method called gradient boosting with SMAP soil moisture in 9 km spatial resolution as a basis. Produced data maps have the same temporal resolution as SMAP (typically daily, but are missing if SMAP soil moisture

retrieval has failed). The developed model is shown to retain the temporal and spatial variability of SMAP soil moisture, but





validated against independent data, the predicted values show better agreement compared to the SMAP soil moisture (RMSE decreasing from 0.097 $m^3/m^3$ to 0.065 $m^3/m^3$, and correlation increasing from 0.30 to 0.52 over forest sites).

Even though SMAP soil moisture has been reported to be sensitive to soil moisture changes under canopy cover (Colliander et al. (2020), Ayres et al. (2021)), the SMAP soil moisture enhancement from 36 km to 9 km spatial resolution has been

reported to cause uncertainty to the soil moisture values (Ambadan et al. (2022)). However, based on comparisons (mean error and standard deviation) between enhanced SMAP soil moisture and in situ values from NF sites in 1 km resolution, the SMAP mission requirement target for soil moisture is achieved, as the mean error (0.03 $m^3/m^3$) is below the target (0.04 $m^3/m^3$, O'Neill et al. (2021)), even though the standard deviation is high (0.12 $m^3/m^3$). Therefore, the enhanced SMAP soil moisture is a suitable basis data for our model. The disadvantage of SMAP soil moisture is that tau-omega-model parameters for each

9 km pixel are based on the dominant land use classification in each pixel. This assumption of homogeneous area ignores the heterogeneous nature of the boreal forest areas (typically a mosaic of forests, peatlands, and water bodies). In the studied area, the SMAP soil moisture is most accurate for forested areas with a herbaceous understory, because, by definition, the dominant land use classification is woody savanna (i.e. a herbaceous understory and forest canopy cover between 30-60%; different class definitions can be found in Strahler et al. (1999)). Therefore, we also have to restrict our model to forested areas. Based on

the land use classification data in 100 m resolution from CORINE land cover for NF (area shown in Figure 1), around 61% of the area is covered by tree cover (42% coniferous trees, 10% broadleaved trees, and 9% mixed trees). One-fifth of the area (almost 20%) is covered in peat bogs, and 4.5% is covered by water bodies (lakes). Water bodies include three large lakes (Lake Inari, Lokka Reservoir, and Porttipahta Reservoir), but also many smaller lakes. The rest of the land use (around 15%) is for different urban activities, grasslands, and agriculture. As our model provides high-resolution soil moisture for forested

areas, it covers approximately 60% of the NF area being covered with this model. Additionally, SMAP soil moisture has a lot of noise, and those features are also transferred to our model predictions. Smoothing would have been one option to decrease the effect of noise, but choosing the method that would have retained the actual temporal variations, was not a straightforward task. Therefore, we decided to leave the noise in the end results.

For modeled soil moisture data, most spatial features are from used soil properties, bulk density, and silt content. Soil

properties are commonly used inputs for soil moisture models (e.g. Ranney et al. (2015), O et al. (2022), Ma et al. (2023), Zhang et al. (2023)), although typically also clay content and sand content are used. As we have an exceedingly small set of individual static inputs in the training set (only 9), we had to exclude clay content and sand content, to prevent overfitting. Additionally, other commonly used inputs include topography and geography data (i.e. elevation, slope, aspect, latitude, and longitude). As we have a very small amount of model construction data, adding geographical information would have caused major overfitting.

We also excluded topography data, as it has been found that models using topography data as inputs may not be useful in other locations (Kemppinen et al. (2023)). Weather-related data, i.e. precipitation, temperature, and solar radiation from ERA5-Land, are included as inputs because they are related to the soil moisture. Precipitation is positively correlated with soil moisture (Sehler et al. (2019)), but air temperature has the opposite effect (Feng and Liu (2015)). Based on feature importances (Figure 2), air temperature has a negative impact on the results as expected, but precipitation does not have the expected positive

effect. This might be due to the coarse resolution of ERA5-Land data (9 km) and the dense vegetation in the area causing soil





moisture to be less sensitive to precipitation. It has also been studied that air temperature has a higher impact on soil moisture than precipitation, even over forest areas (Feng and Liu (2015)). This effect can be seen in feature importances (Figure 2), as temperature has a clearly higher impact on the soil moisture estimates compared to precipitation. Solar radiation impacts soil moisture values indirectly through evapotranspiration (Liu et al. (2012)). The cumulated precipitation and temperature values

(a sum over 72 hours) increased the model accuracy compared to the instantaneous values, and therefore they were chosen. The solar radiation had the opposite effect and hence we chose the instantaneous values. An additional useful data source would have been the land surface temperature (LST), as the LST difference between night and day correlates with soil moisture. LST data has been widely used for estimating soil moisture (e.g. Matsushima et al. (2012), Hao et al. (2022), Han et al. (2023)). The disadvantage of LST is that it is obtained from optical measurements. Due to the difficulties caused by cloud cover, obtaining

even moderately gap-free LST data regularly over the whole NF area was an impossible task, and therefore we did not include LST as an input.

To choose the best model, we tested three different tree-based methods: random forest, level-wise gradient boosting, and leaf-wise growth-based gradient boosting. The leaf-wise growth GB (lightGBM) produced the results with the highest accuracy and was therefore chosen. However, because it is a tree-based method, it cannot extrapolate well when inputs differ from training

data, as the decision boundaries are determined during training. Therefore, our predicted estimates are more or less bounded, and unexpectedly high or low soil moisture values are not predicted correctly. During the model construction process, we discovered that model construction is highly sensitive to changes in static inputs due to the small study data set. Therefore, the data division into training and test sets was not random but carefully considered. Almost all in situ sites are located in forested areas, a fact that further strengthened the model's usefulness for only forested areas.

We included SMAP in 1 km resolution to be compared to our model predictions. There exist other downscaled data sets, but as the soil moisture network is sparse, they, unfortunately, use some of the same in situ sites from NF and Alaska for training, and therefore we couldn't use them as independent data sets. The comparisons with SMAP in 1 km resolution indicate that downscaled SMAP lacks some of the variability found in our model. For some sites in Alaska, SMAP in 1 km even performed weaker compared to SMAP in 9 km resolution. Based on those results, it could be possible that thermal inertia theory is not

ideal for downscaling soil moisture data over forested areas.

In the future, L-band-based missions, like the NASA-ISRO SAR mission (NISAR, https://nisar.jpl.nasa.gov/) with a planned launch at the end of 2024, and Radar Observing System for Europe in L-band (ROSE-L, https://sentiwiki.copernicus.eu/web/rose-l) with a planned launch in 2028, are aiming to provide soil moisture data with higher spatial resolution (around 10 m for NISAR and around 25 m for ROSE-L). With those resolutions, even peatlands can be taken into account. As over 20% of the NF area

is covered by peats and bogs, and considering their importance as carbon sinks, peatlands need to be included in soil moisture studies. If there were more in situ observation sites located in peatlands, one could construct a model based on them, and then combine models focused on forested areas and peatlands to better account for all the variability in soil moisture over boreal forest areas.



## 6 Conclusions

We developed a model to predict high-resolution soil moisture values in boreal forests. This model specifically targets forested regions, as peatlands are not represented in SMAP soil moisture data, and most in situ soil moisture observation sites are located in forested areas. The model is developed by using SMAP soil moisture at 9 km spatial resolution as the basis data, and additional vegetation and soil properties are used to guide the machine learning model together with in situ soil moisture values. The model produces predictions at a resolution of 1 km, which aligns well with SMAP measurements. However, it

can also generate soil moisture estimates at a finer resolution of 250 m, offering improved accuracy in certain applications, for example hydrological modelling and carbon exchange studies. Consequently, this model provides a valuable tool for predicting soil moisture in high resolution across boreal forested landscapes.

*Data availability.* The data sets associated with this paper are available in the Finnish Meteorological Institute Research Data repository METIS (http://hdl.handle.net/11304/e8183441-bb2a-4866-a07e-2996b022191a).





**Table 1.** Gradient Boosting model parameters, their ranges and chosen values to be used for model building. Parameter ranges are constrained to prevent overfitting. The chosen values are determined by using GridSeachCV method with CV=3.

| Parameter name | Range | Chosen value |
| --- | --- | --- |
| Number of leaves | [3,4,5,6] | 5 |
| Maximum depth | [2,3,4,5] | 4 |
| Minimal amount of data in one leaf | [10,20,30,40] | 20 |
| Maximum number of bins | [30,40,50,60,70] | 60 |





**Table 2.** Statistical values between predicted values and in situ soil moisture for training and test sets.

|  | RMSE | URMSE | R | $R^2$ | N |
|---|---|---|---|---|---|
| Training | 0.064 | 0.064 | 0.86 | 0.74 | 4480 |
| Test all | 0.078 | 0.063 | 0.40 | 0.16 | 2221 |
| Test (DIS0004 excluded) | 0.049 | 0.045 | 0.66 | 0.44 | 2221 |
| Test site A | 0.046 | 0.043 | 0.64 | 0.41 | 534 |
| Test site B | 0.070 | 0.049 | 0.72 | 0.52 | 579 |
| Test site C | 0.040 | 0.034 | 0.71 | 0.50 | 509 |
| Test site D | 0.120 | 0.023 | 0.77 | 0.59 | 599 |
| Test site D (DIS0004 excluded) | 0.034 | 0.025 | 0.72 | 0.52 | 599 |





**Table 3.** Sensitivities for chosen inputs, standard deviation between test set in situ soil moisture and predicted soil moisture, and calculated total uncertainty of the model. All results have the unit $m^3/m^3$.

| source | uncertainty |
| --- | --- |
| Bulk density | 0.057 |
| Silt content | 0.075 |
| SMAP soil moisture | 0.035 |
| NDVI | 0.003 |
| EVI | 0.004 |
| Test standard deviation | 0.065 |
| total | 0.119 |





**Table 4.** Model validation statistics between observed in situ soil moisture from Alaska sites and predicted soil moisture values.

| RMSE | forest | mosaic | shrub | sparse | all |
|---|---|---|---|---|---|
| SMAP | 0.097 | **0.122** | 0.193 | 0.112 | 0.137 |
| SMAP 1 km | 0.112 | 0.136 | **0.170** | 0.123 | **0.133** |
| Predicted 1 km | 0.067 | 0.152 | 0.182 | **0.110** | 0.137 |
| Predicted 250 m | **0.065** | 0.138 | 0.185 | **0.110** | 0.134 |
| **URMSE** | | | | | |
| SMAP | 0.096 | 0.121 | 0.134 | **0.104** | 0.133 |
| SMAP 1 km | 0.097 | 0.131 | 0.164 | 0.108 | 0.128 |
| Predicted 1 km | 0.067 | 0.119 | 0.114 | 0.106 | 0.118 |
| Predicted 250 m | **0.065** | **0.117** | **0.106** | 0.108 | **0.113** |
| **R** | | | | | |
| SMAP | 0.30 | 0.37 | -0.17 | 0.06 | 0.18 |
| SMAP 1 km | 0.02 | 0.13 | -0.65 | 0.00 | 0.12 |
| Predicted 1 km | 0.47 | 0.54 | -0.14 | -0.01 | **0.22** |
| Predicted 250 m | **0.52** | **0.58** | -0.01 | -0.02 | 0.20 |
| **N** | | | | | |
| SMAP | 418 | 402 | 439 | 367 | 1626 |
| SMAP 1 km | 434 | 407 | 240 | 374 | 1455 |
| Predicted 1 km | 497 | 484 | 548 | 421 | 1950 |
| Predicted 250 m | 405 | 288 | 409 | 308 | 1432 |





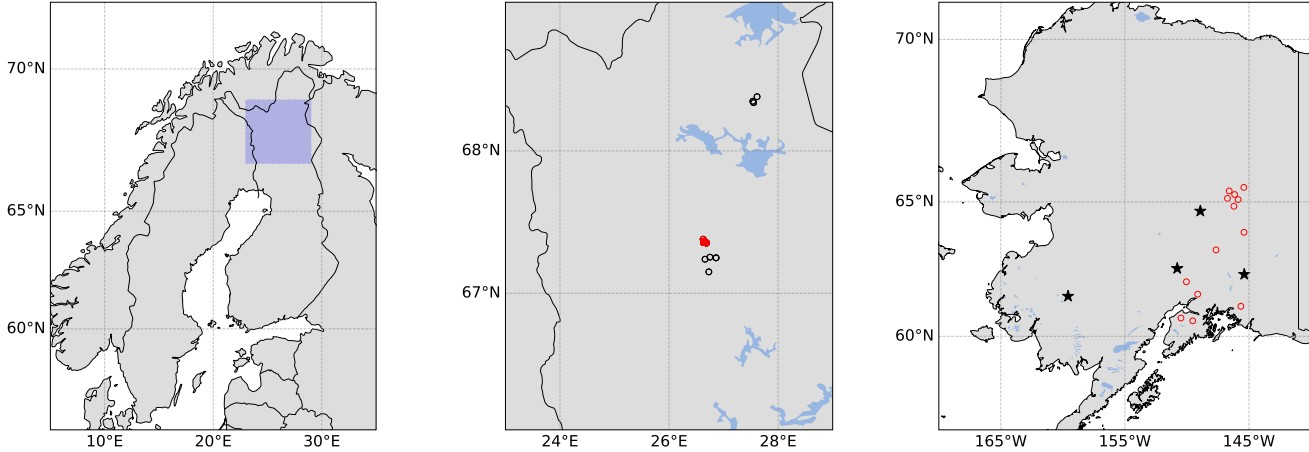

**Figure 1.** Locations of the chosen training, test, and validation data sites. Left: Northern Finland study area in a broader context (blue squared area). Middle: Location of the chosen model training (black circles) and test (red circles) in situ sites. Right: Location of the chosen model validation in situ sites. Black stars indicate forest sites and red circles indicate other sites (mosaic, shrub, sparse).





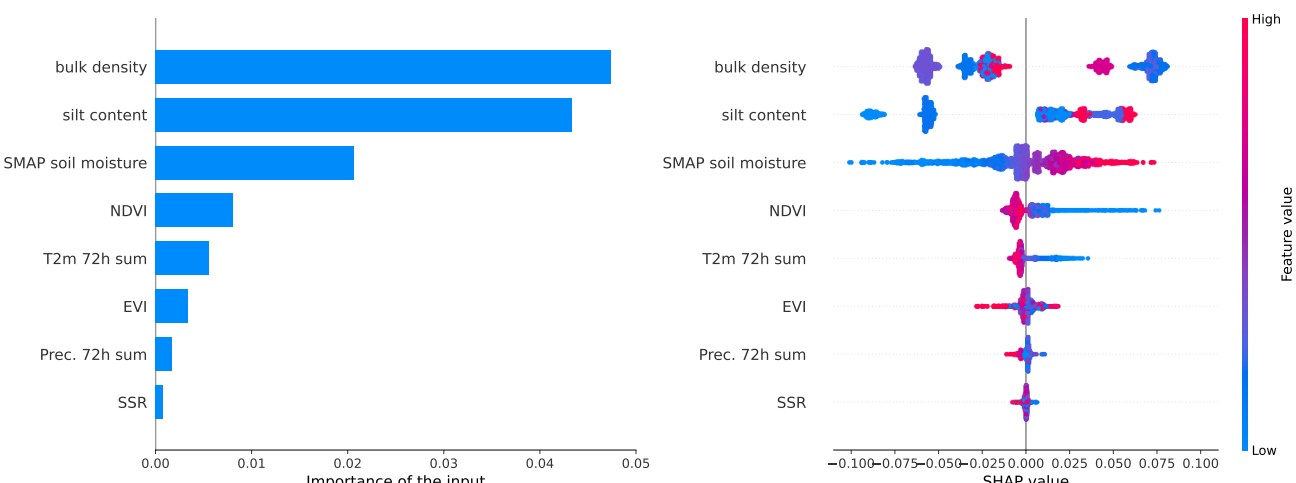

**Figure 2.** The SHapley Additive exPlanations (SHAP) values for the constructed gradient boosting model. Left: the mean SHAP values for each predictor. Right: More detailed view of the effect of different feature values on predictions.



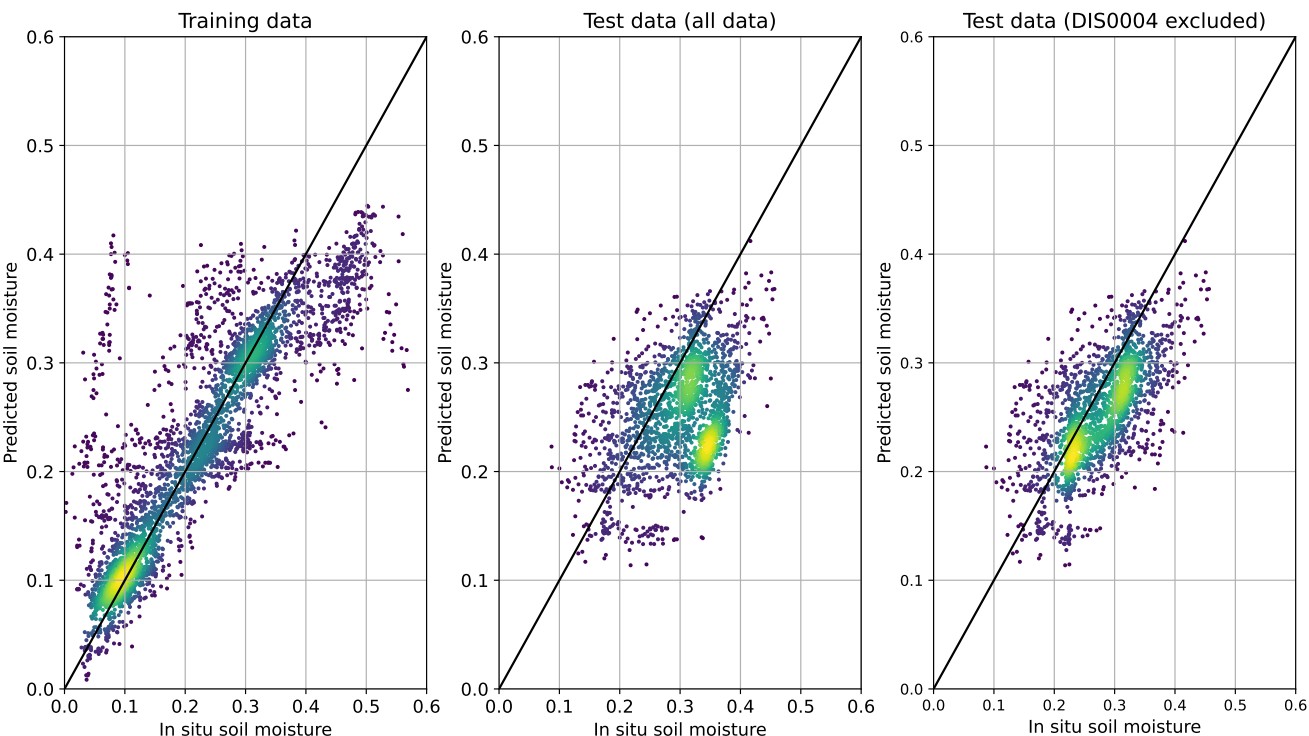

**Figure 3.** Scatter plots of predicted training and test set soil moisture values from years 2019–2023. Left: scatter plot of training data set. Middle: scatter plot of test set. Right: scatter plot of test set, where one site (DIS0004) is removed from the set as site's soil moisture values deviated clearly from the SMAP soil moisture values.





**Figure 4.** Exemplary time series of test sites for the year 2023. Predicted soil moisture values in 1 km and 250 m resolutions are from a developed gradient boosting model.



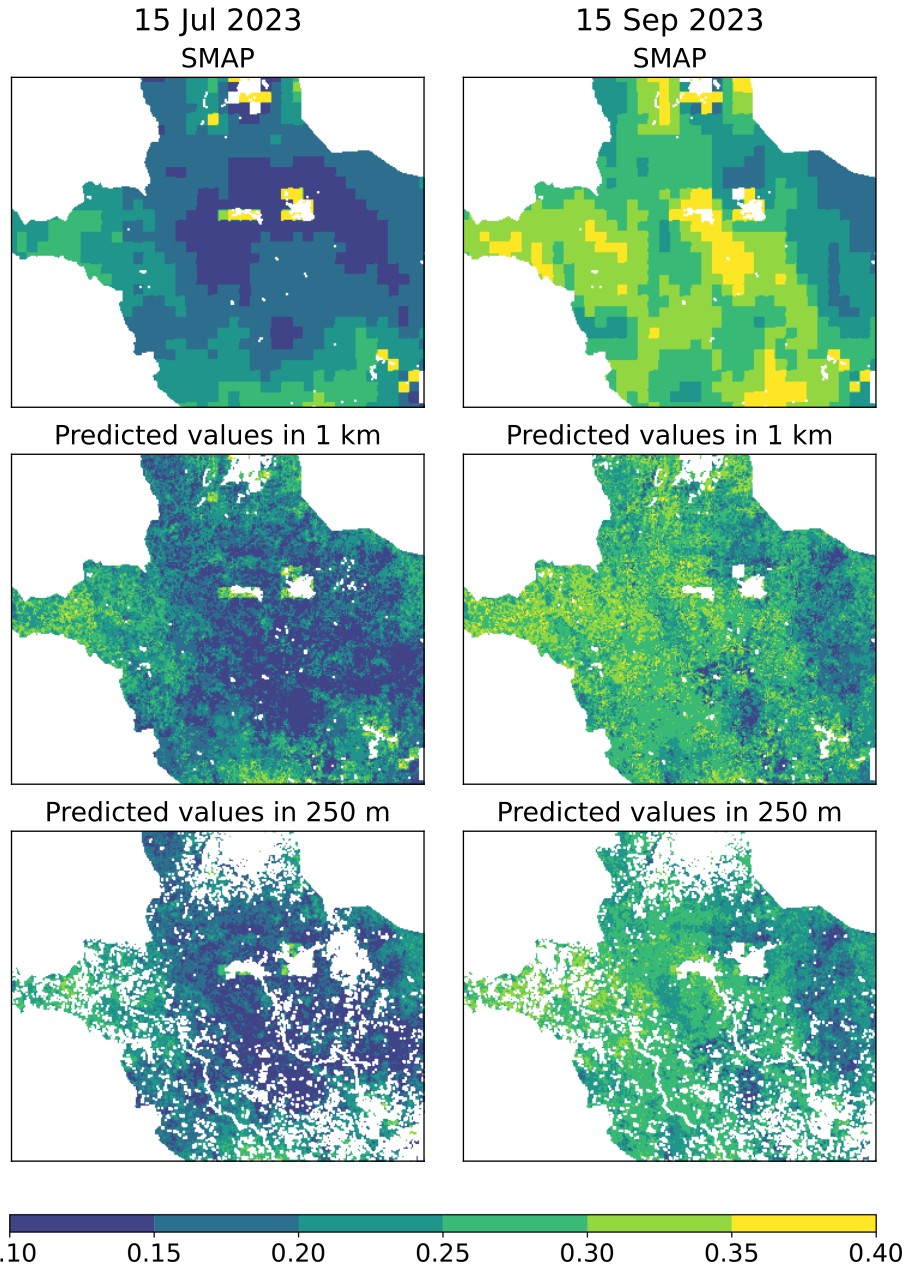

**Figure 5.** Exemplary maps show SMAP soil moisture for two dates, along with predicted soil moisture at spatial resolutions of 1 km and 250 m. Missing values due to the missing values in inputs and water bodies are indicated with white. Even though developed model is just for forested areas, all pixels with data in these maps are shown for clarity.

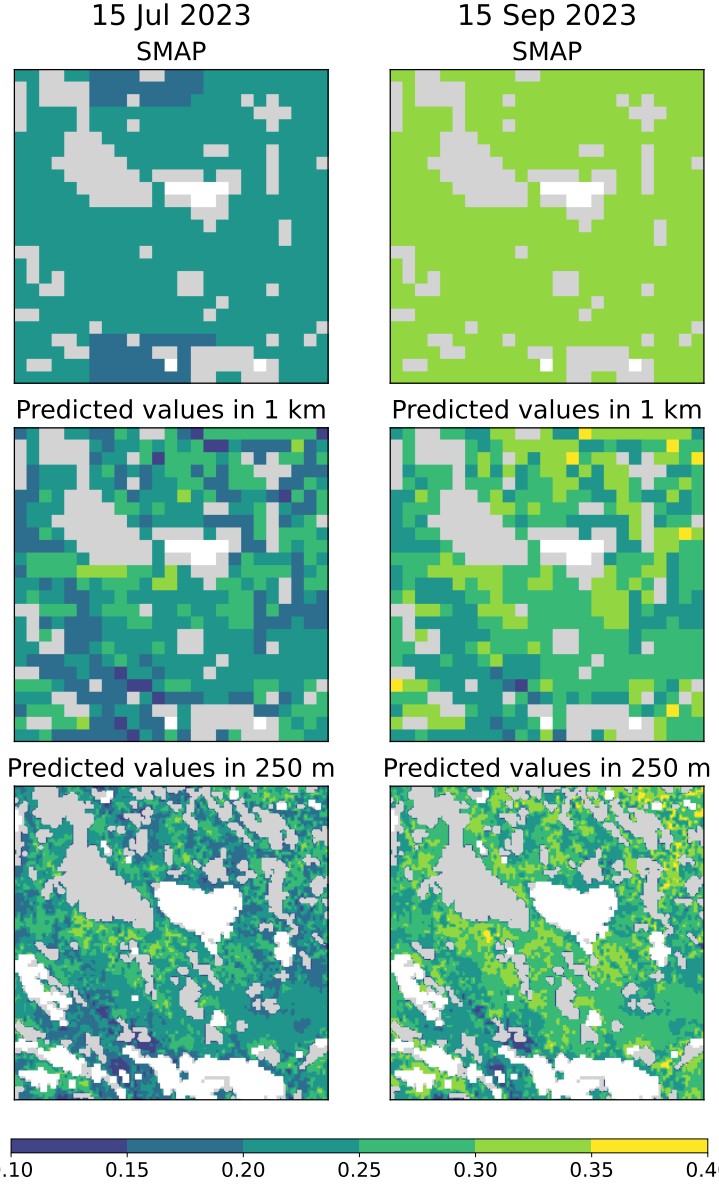

**Figure 6.** Exemplary maps show SMAP soil moisture for two dates, along with predicted soil moisture at spatial resolutions of 1 km and 250 for smaller area located around Lake Pallas (N68.033°, E24.197°). Missing values due to the water bodies are indicated with white and other land uses than forest are indicated with grey. The land use mask is based on CORINE land use classification in 100 m resolution. Pixel is assumed to be forest if the forest class fraction is above 50 %.



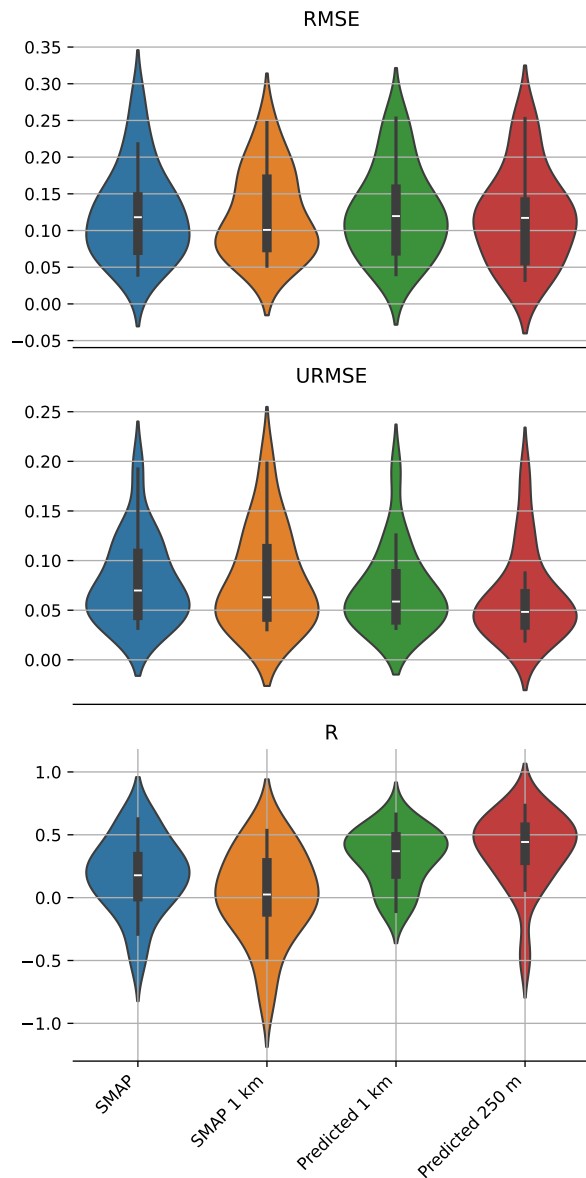

**Figure 7.** RMSE, URMSE, and R values between in situ values from Alaska validation data set and different soil moisture data sets (SMAP in 9 km resolution, SMAP in 1 km resolution, predicted values using 1 km resolution data, and 250 m resolution data) shown as violin plots. The data is from the time period between 1 May 2019 and 15 Oct 2019.



**Figure 8.** Comparisons between in situ values from Alaska sites (four forested sites) and different soil moisture data sets (SMAP in 9 km resolution, SMAP in 1 km resolution, predicted values using 1 km resolution data, and 250 m resolution data). The data is from the period between May 1, 2019, and October 15, 2019.

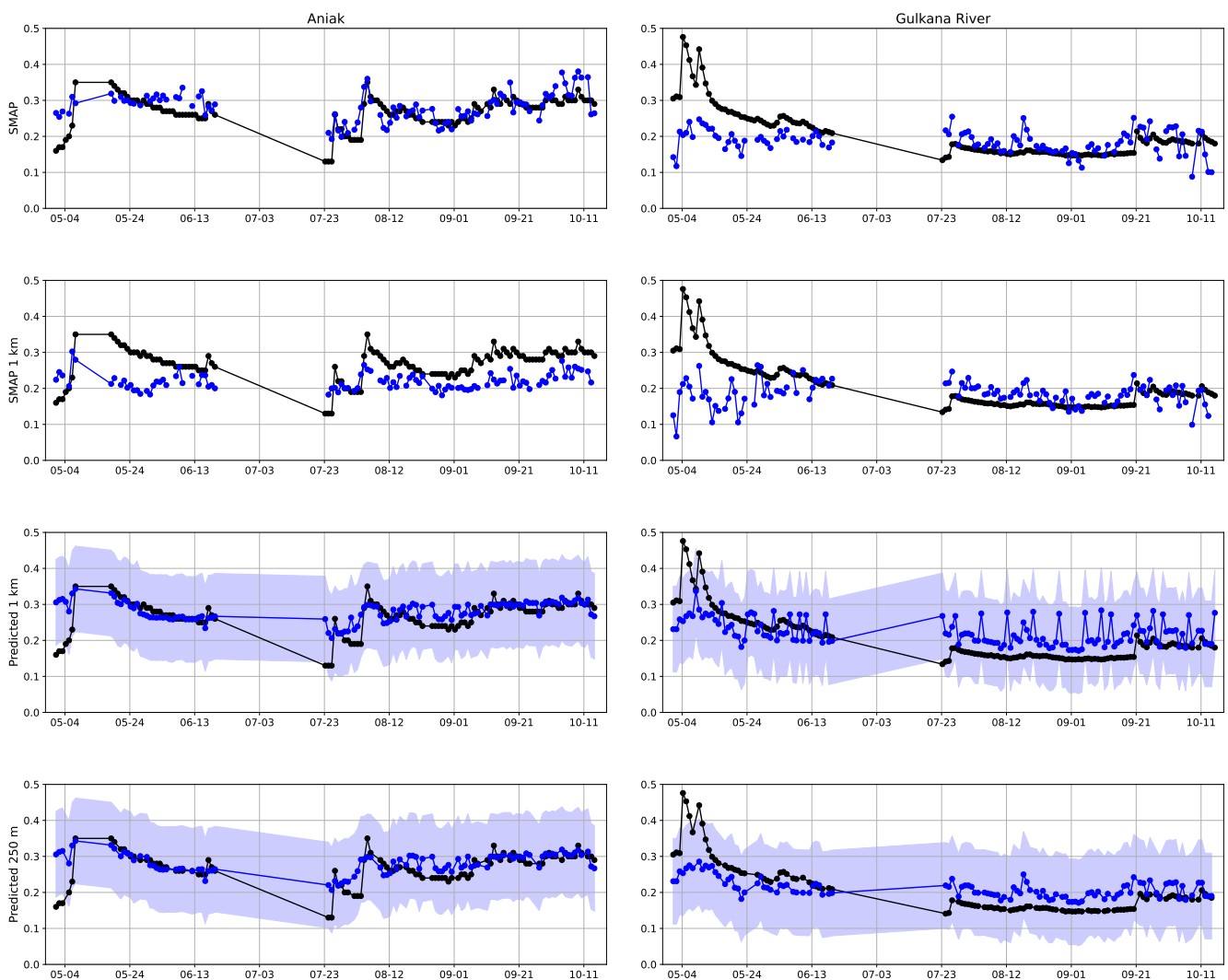

**Figure 9.** Exemplary time series for two Alaska sites located in forested areas. Left: Aniak, right: Gulkana River. Black indicates in situ soil moisture and blue satellite-based soil moisture data. The curtain in the two bottom rows indicates the model uncertainty (uncertainty 0.119 $m^3/m^3$ added and subtstracted from the predicted values). All data are from the year 2019.





**Appendix A**

**A1**

**Table A1.** In situ sites for training and testing the soil moisture model, located in Northern Finland. Land cover information is from CORINE land cover data set.

| Name | Number of spots | Location | Land use | Bulk density (cg/cm$^3$) | Silt content (g/kg) |
|---|---|---|---|---|---|
| Training set | | | | | |
| IOA0002 | 1 | N67.362 , E26.634 | Coniferous forest | 0.585 | 29.32 |
| IOA0003[a] | 7 | N67.362 , E26.634 | Coniferous forest | 0.600 | 28.68 |
| IOA0007 | 2 | N67.362 , E26.634 | Coniferous forest | 0.585 | 29.32 |
| KAI0001 | 3 | N67.357 , E26.685 | Coniferous forest | 0.592 | 27.81 |
| KAI0002[b] | 2 | N67.359 , E26.686 | Coniferous forest | 0.571 | 27.92 |
| LEN0001 | 2 | N67.384 , E26.625 | Coniferous forest | 0.596 | 28.99 |
| MET0003 | 1 | N67.362 , E26.637 | Coniferous forest | 0.584 | 27.70 |
| MET0010 | 1 | N67.362 , E26.638 | Coniferous forest | 0.584 | 27.70 |
| SAA0001 | 3 | N68.330 , E27.550 | Moors and heathland | 0.630 | 34.54 |
| SAA0002 | 2 | N68.339 , E27.535 | Transitional woodland-shrub | 0.654 | 34.44 |
| SAA0003 | 2 | N68.370 , E27.614 | Coniferous forest | 0.641 | 31.48 |
| SUO0004 | 2 | N67.367 , E26.651 | Peats and bogs | 0.616 | 28.31 |
| SUO0005 | 2 | N67.370 , E26.640 | Coniferous forest | 0.585 | 29.32 |
| Test set | | | | | |
| DIS0001[c] | 3 | N67.257 , E26.749 | Coniferous forest | 0.557 | 29.37 |
| DIS0002[d] | 3 | N67.153 , E26.729 | Transitional woodland-shrub | 0.563 | 29.18 |
| DIS0003[e] | 2 | N67.243 , E26.660 | Coniferous forest | 0.629 | 30.53 |
| DIS0004[f] | 2 | N67.253 , E26.862 | Coniferous forest | 0.551 | 27.79 |
| DIS0005[f] | 1 | N67.253 , E26.861 | Coniferous forest | 0.551 | 27.79 |

[a] OA0003 spot 4 locates in a different pixel (bulk density 0.585 cg/cm$^3$, silt content 29.32g/kg).

[b] KAI0002 has 3 spots, but one of them had abnormally low soil moisture values and was therefore removed.

[c] Test site A.

[d] Test site B.

[e] Test site C.

[f] Test site D.





**Table A2.** In situ sites for model validation, sites located in Alaska. Data are from three different networks, SCAN (Schaefer et al. (2007)), SNOTEL (Leavesley et al. (2008), Leavesley (2010)), and USCRN (Bell et al. (2013)). Land cover information is from ESA CCI Land Cover (ESA (2017)).

| Name | Location | Network | Land use |
| --- | --- | --- | --- |
| Aniak | N61.58, W159.58 | SCAN | Tree cover |
| Eagle Summit | N65.49, W145.41 | SNOTEL | Sparse vegetation (tree shrub herbaceous cover) (<15%) |
| Granite Creek | N63.94, W145.40 | SNOTEL | Sparse vegetation (tree shrub herbaceous cover) (<15%) |
| Gulkana River | N62.41, W145.38 | SCAN | Tree cover |
| Kenai 29 Ene | N60.73, W150.45 | USCRN | Shrub or herbaceous cover flooded fresh/saline/brakish water |
| Kenai Moose Pens | N60.73, W150.48 | SCAN | Shrub or herbaceous cover flooded fresh/saline/brakish water |
| Little Chena Ridge | N65.12, W146.73 | SNOTEL | Mosaic tree and shrub (>50%) / herbaceous cover (<50%) |
| Monahan Flat | N63.31, W147.65 | SNOTEL | Mosaic tree and shrub (>50%) / herbaceous cover (<50%) |
| Monument Creek | N65.08, W145.87 | SNOTEL | Mosaic tree and shrub (>50%) / herbaceous cover (<50%) |
| Mt. Ryan | N65.25, W146.15 | SNOTEL | Mosaic tree and shrub (>50%) / herbaceous cover (<50%) |
| Munson Ridge | N64.85, W146.21 | SNOTEL | Sparse vegetation (tree shrub herbaceous cover) (<15%) |
| Nenana | N64.68, W148.92 | SCAN | Tree cover |
| Summit Creek | N60.62, W149.53 | SNOTEL | Shrubland |
| Susitna Valley High | N62.13, W150.04 | SNOTEL | Shrub or herbaceous cover flooded fresh/saline/brakish water |
| Spring Creek | N61.66, W149.13 | SCAN | Shrub or herbaceous cover |
| Tokositna Valley | N62.63, W150.78 | SNOTEL | Tree cover mixed leaf type (broadleaved and needleleaved) |
| Upper None Creek | N65.37, W146.59 | SNOTEL | Sparse vegetation (tree shrub herbaceous cover) (<15%) |
| Upper Tsaina River | N61.19, W145.65 | SNOTEL | Mosaic tree and shrub (>50%) / herbaceous cover (<50%) |



*Author contributions.* TV and MA are responsible for acquiring the funding for the research presented in this paper. TV and EJ are behind the conceptualization of research questions presented in this paper. EJ collected and processed all the data used in this paper. EJ developed methodology for analyses, carried them out and investigated the obtained results. EJ also led the preparation of manuscript, with contributions
from all co-authors. All authors reviewed and edited the manuscript.

*Competing interests.* The contact author has declared that none of the authors has any competing interests.

*Acknowledgements.* This work was financially supported by the Research Council of Finland in the project CWI (347662). We acknowledge support from the Ministry of Transport and Communications through the Integrated Carbon Observing System (ICOS) Finland. The authors would like to thank the National Snow and Ice Data Center, Copernicus Data Space Ecosystem, and NASA for providing the satellite data
used in this study, ISRIC for providing the soil properties, and the Finnish Meteorological Institute and the International Soil Moisture Network for providing the in situ measurements used in this study. Furthermore, this publication has been prepared using European Union's Copernicus Land Monitoring Service information (https://doi.org/10.2909/960998c1-1870-4e82-8051-6485205ebbac). The authors would like to thank Aku Riihelä, Kimmo Rautiainen, Niilo Kalakoski, and Jaakko Ikonen (Finnish Meteorological Institute) for helpful discussions.



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
