# Peer review of "High-resolution soil moisture mapping in northern boreal forests using SMAP data and downscaling techniques"

_Hydrology and Earth System Sciences, 2024_

## Author Comment (AC1)

We thank reviewer #1 for their comments on our manuscript. Our answers to the comments are below in blue, after the reviewer's comments.

The manuscript presents a machine-learning-based approach to downscale SMAP soil moisture data from 9 km to finer resolutions of 1 km and 250 m for boreal forests. The model integrates SMAP data with soil, vegetation, and weather inputs to provide higher spatial resolution soil moisture estimates, addressing the limitations of SMAP's coarse coverage in northern latitudes. Validation against in situ measurements shows improved accuracy, with reduced RMSE and increased correlation compared to raw SMAP data. However, the methodology is limited to forested areas, excluding peatlands and other land types. While the approach demonstrates the potential for high-resolution soil moisture mapping, several areas require substantial improvement before publication.

**Major Comments:**

1. SMAP Mission provides SMAP-Sentinel 3 km and 1 km soil moisture (https://doi.org/10.1016/j.rse.2019.111380), and it is very strange to see that these are not discussed in the literature section.
   - We apologize; this was an oversight on our part. We will add a discussion about SMAP/Sentinel data to our manuscript.

2. One of the key advantages of this study is its complement to the SMAP Sentinel Soil Moisture product, particularly by addressing NASA's limitation in providing soil moisture data over northern latitudes. However, while this contribution is acknowledged, the paper could have been strengthened significantly by demonstrating a more direct comparison with SMAP Sentinel dense time series in areas where such data are available. I would suggest the author replicate the same method over the mainland where SMAP Sentinel retrieval is available and compare for multiple locations. This can help a wider audience understand how the discussed method is reliable when compared to the operational product. This would provide a robust validation framework and establish the superiority or limitations of the proposed methodology.
   - We acknowledge this point; however, the model constructed in our manuscript is meant only for boreal forested areas. We consider that this kind of additional model construction and data compilation is out of the scope of this study and would need another paper.

3. The reliance on static inputs such as bulk density and silt content raises concerns about the adaptability of the model to regions beyond the boreal forests of Northern Finland. The training set's limited geographic and environmental variability suggests that the model may not perform well in regions with differing soil or vegetation characteristics. This could potentially undermine the generalizability of the approach, and expanding the training dataset to include diverse boreal forest sites would address this shortcoming.

- Unfortunately, the number of freely available boreal forest soil moisture in situ sites and their data is very limited, which prevents us from adding more in situ sites with different soil properties. Therefore, to demonstrate the applicability of the constructed model beyond Northern Finland, we validated it against soil moisture data from forested sites in Alaska. These sites differ notably in soil properties compared to the Finnish dataset: in Northern Finland's sites the bulk density ranges from 0.55 cg/cm$^3$ to 0.65 cg/cm$^3$ and silt content from 27.7 g/kg to 34.5 g/kg while in Alaskan sites bulk density varies from 0.41 cg/cm$^3$ to 0.79 cg/cm$^3$ and silt content from 48.9 g/kg to 70.2 g/kg. We acknowledge that this point should have been stated more clearly in the manuscript. We will revise the text to better explain the choices and clarify the limitations related to data availability.

4. The exclusion of peatlands from the study is a significant limitation, especially given their critical role in carbon storage in boreal ecosystems. Although the authors briefly discuss this gap, they fail to propose a concrete pathway for integrating peatlands into future models. More effort should have been made to outline how the methodology could be adapted to incorporate such essential land cover types.

    - Peatlands are excluded as there is data from only a few peatlands in situ sites available. As peatland can vary from dry to almost saturated, there would need to be many more in situ sites to be able to add peatland values to the model. In addition, there is great variation in moisture conditions at such a small spatial scale that even our 250 m pixel size is insufficient to capture it realistically. We have already outlined in the manuscript how peatlands could be included (L. 343-345).

5. The discussion around uncertainty analysis highlights the model's heavy dependence on soil properties, which dominate the prediction outcomes. While these are undoubtedly critical inputs, the relative insensitivity of the model to weather-related inputs like precipitation suggests a potential flaw in the approach. The coarse resolution of ERA5-Land data used for precipitation might be a

contributing factor, and exploring higher-resolution meteorological datasets could refine the model's sensitivity to short-term climatic variations.

- Yes, this is a good point. We will examine whether replacing ERA5-L data with higher spatial resolution precipitation and temperature data (e.g. 1 km spatial resolution data from https://en.ilmatieteenlaitos.fi/gridded-observations-on-aws-s3) improves the results. If that is the case, we will then explore the possibility of using a similar approach (i.e. kriging interpolation to weather station data) to provide precipitation and other quantities on around 1 km spatial resolution over Alaska (as this kind of data set does not, to our knowledge, exist). We will also explore the possibility of using satellite-based data to replace ERA5-L data.

6. The use of a machine-learning-based gradient boosting model (LightGBM) is appropriate for capturing complex relationships, but the small training dataset limits the robustness of the approach. Therefore, it's important to discuss the limitations of the method used and how to overcome them.

- We will add more text to the manuscript about this.

7. The SMAP L3_SM_P_E spatial resolution is 33 km, which is gridded to 9 km, but this is not mentioned anywhere in the manuscript. Typically, downscaling should consider the original 33 km resolution rather than the 9 km gridded resolution. If the model directly uses the 9 km gridded data as the spatial resolution, I recommend reprocessing the model by considering the original 33 km resolution. Additionally, the revised version should clearly explain how this resolution is incorporated into the methodology.

- The original resolution of SMAP L3_SM_P_E and the gridded one are mentioned in the manuscript in L. 35—36, and L. 69. It is true, that downscaling is typically done by using the original resolution, but not always. For example, the SMAP downscaled to 1 km product (Fang et al. 2022) uses the enhanced 9 km data product. Also, as we do have only a few in situ stations available, using the original spatial resolution would lead to even fewer static parameter values. Therefore, we cannot change the SMAP data to coarser resolution data.

Fang, B., Lakshmi, V., Cosh, M., Liu, P.-W., Bindlish, R., and Jackson, T. J.: A global 1-km downscaled SMAP soil moisture product based on thermal inertia theory, Vadose Zone Journal, 21, e20 182, https://doi.org/https://doi.org/10.1002/vzj2.20182, 2022

8. While the authors discuss future L-band missions like NISAR and ROSE-L, more focus should be given how this method will be useful for this mission.
   - We will add text about this to the manuscript.

9. Validation against in situ measurements shows promising accuracy improvements, but the exclusion of outliers like DIS0004 suggests sensitivity to anomalies that the model should handle better.
   - We will look into this as well. It is possible that the site DIS0004 actually has very different silt content (and also maybe bulk density) values than what we acquire from SoilGrids, which could explain the outlier nature of the DIS0004.

10. The conclusion section needs attention as it does not read well. Please consider rewriting the section more scientifically.
    - We will consider this.

**Minor Comments:**

1. L2: "Phenomena" is not appropriate here.
   - We will correct that.
2. L4: "High spatio-temporal scale" would be more appropriate.
   - We will change "high temporal and spatial scales" to "High spatio-temporal scale"
3. L37: "Short distance" is not appropriate. Rephrase as "spatially heterogeneous."
   - We will correct this.
4. L41: Citation of the operational 1 km SMAP soil moisture product is missing (https://doi.org/10.1016/j.rse.2019.111380).
   - We will add this citation.
5. There are multiple sites in Alaska where in-situ soil moisture is available, and those should be included, such as the site from Delta Junction (NEON site).
   - We will add Delta Junction site to the validation.
6. A description of the study site is required in the main text.
   - We will add the description of study site to the manuscript.
7. It is unclear why CORINE land cover is used. The new ESA 10 m land cover provides more sufficient information for this study and spatially has more detail than CORINE.

- CORINE has a spatial resolution of 100 m, which is closer to the smallest acquired spatial resolution of 250 m. ESA 10m WorldCover data has unnecessarily high spatial resolution for our study.

8. L336: The NISAR mission will provide a 200 m soil moisture product as an operational soil moisture product. It is suggested to include proper citations in the manuscript (https://doi.org/10.1016/j.rse.2023.113667 and https://doi.org/10.1016/j.rse.2024.114288).

    - We will add these citations to the manuscript.

9. L338: NISAR will be launched in April 2025.

    - We will correct this.

---

## Author Response (AR1)

**Answers to reviewer #1**

We thank reviewer #1 for their comments on our manuscript. Our answers to the comments are below in blue, after the reviewer's comments.

The manuscript presents a machine-learning-based approach to downscale SMAP soil moisture data from 9 km to finer resolutions of 1 km and 250 m for boreal forests. The model integrates SMAP data with soil, vegetation, and weather inputs to provide higher spatial resolution soil moisture estimates, addressing the limitations of SMAP's coarse coverage in northern latitudes. Validation against in situ measurements shows improved accuracy, with reduced RMSE and increased correlation compared to raw SMAP data. However, the methodology is limited to forested areas, excluding peatlands and other land types. While the approach demonstrates the potential for high-resolution soil moisture mapping, several areas require substantial improvement before publication.

**Major Comments:**

- 1. SMAP Mission provides SMAP-Sentinel 3 km and 1 km soil moisture (<a href="https://doi.org/10.1016/j.rse.2019.111380">https://doi.org/10.1016/j.rse.2019.111380</a>), and it is very strange to see that these are not discussed in the literature section.
  - We apologize; this was an oversight on our part. We added text about SMAP/Sentinel data to our manuscript (L34-41).
- 2. One of the key advantages of this study is its complement to the SMAP Sentinel Soil Moisture product, particularly by addressing NASA's limitation in providing soil moisture data over northern latitudes. However, while this contribution is acknowledged, the paper could have been strengthened significantly by demonstrating a more direct comparison with SMAP Sentinel dense time series in areas where such data are available. I would suggest the author replicate the same method over the mainland where SMAP Sentinel retrieval is available and compare for multiple locations. This can help a wider audience understand how the discussed method is reliable when compared to the operational product. This would provide a robust validation framework and establish the superiority or limitations of the proposed methodology.
  - We acknowledge this point; however, the model constructed in our manuscript is meant only for boreal forested areas. We consider that this kind of additional model construction and data compilation is out of the scope of this study and would need another paper.

- 3. The reliance on static inputs such as bulk density and silt content raises concerns about the adaptability of the model to regions beyond the boreal forests of Northern Finland. The training set's limited geographic and environmental variability suggests that the model may not perform well in regions with differing soil or vegetation characteristics. This could potentially undermine the generalizability of the approach, and expanding the training dataset to include diverse boreal forest sites would address this shortcoming.
  - Unfortunately, the number of freely available boreal forest soil moisture in situ sites and their data is very limited, which prevents us from adding more in situ sites with different soil properties. Due to that, and also due to the other changes to the input data, we decided to remove soil properties information altogether.
- 4. The exclusion of peatlands from the study is a significant limitation, especially given their critical role in carbon storage in boreal ecosystems. Although the authors briefly discuss this gap, they fail to propose a concrete pathway for integrating peatlands into future models. More effort should have been made to outline how the methodology could be adapted to incorporate such essential land cover types.
  - Peatlands are excluded as there is data from only a few peatlands in situ sites available. As peatland can vary from dry to almost saturated, there would need to be many more in situ sites to be able to add peatland values to the model. In addition, there is great variation in moisture conditions at such a small spatial scale that even our 250 m pixel size is insufficient to capture it realistically. We have already outlined in the manuscript how peatlands could be included (L. 370-372).
- 5. The discussion around uncertainty analysis highlights the model's heavy dependence on soil properties, which dominate the prediction outcomes. While these are undoubtedly critical inputs, the relative insensitivity of the model to weather-related inputs like precipitation suggests a potential flaw in the approach. The coarse resolution of ERA5-Land data used for precipitation might be a contributing factor, and exploring higher-resolution meteorological datasets could refine the model's sensitivity to short-term climatic variations.
  - Yes, this is a good point. We replaced ERA5-L data with higher spatial resolution precipitation and temperature data (e.g. 1 km spatial resolution data from <a href="https://en.ilmatieteenlaitos.fi/gridded-observations-on-aws-s3">https://en.ilmatieteenlaitos.fi/gridded-observations-on-aws-s3</a>), which did

improve the results. Unfortunately, a similar approach was not possible over Alaska, and therefore ERA5-Land data was used there for temperature data. For precipitation, we used Global Precipitation Measurement mission data. We have modified the text due to these changes (Subsections 2.5, 2.6, 2.7; L. 174-177, L. 264-268; L.328-340)

- 6. The use of a machine-learning-based gradient boosting model (LightGBM) is appropriate for capturing complex relationships, but the small training dataset limits the robustness of the approach. Therefore, it's important to discuss the limitations of the method used and how to overcome them.
  - We have added more text to the manuscript about this (L. 350-352).
- 7. The SMAP L3\_SM\_P\_E spatial resolution is 33 km, which is gridded to 9 km, but this is not mentioned anywhere in the manuscript. Typically, downscaling should consider the original 33 km resolution rather than the 9 km gridded resolution. If the model directly uses the 9 km gridded data as the spatial resolution, I recommend reprocessing the model by considering the original 33 km resolution. Additionally, the revised version should clearly explain how this resolution is incorporated into the methodology.
  - This is true. As we were not sure how to implement the original 33 km resolution information, we decided to change the SMAP at 9 km resolution to the SMAP at 36 km resolution. This led to major rewritings and changes to the model, data sets, and results (e.g. subsection 3.2, section 4)
- 8. While the authors discuss future L-band missions like NISAR and ROSE-L, more focus should be given how this method will be useful for this mission.
  - We added text about this to the manuscript (L. 372-373).
- 9. Validation against in situ measurements shows promising accuracy improvements, but the exclusion of outliers like DIS0004 suggests sensitivity to anomalies that the model should handle better.
  - We investigated this, and it was an error on soil properties on our behalf.
- 10. The conclusion section needs attention as it does not read well. Please consider rewriting the section more scientifically.
  - The conclusions section has been rewritten in some parts.

**Minor Comments:**

- 1. L2: "Phenomena" is not appropriate here.
  - We corrected that.
- 2. L4: "High spatio-temporal scale" would be more appropriate.
  - We changed "high temporal and spatial scales" to "High spatio-temporal scale"
- 3. L37: "Short distance" is not appropriate. Rephrase as "spatially heterogeneous."
  - We corrected this.
- 4. L41: Citation of the operational 1 km SMAP soil moisture product is missing (<a href="https://doi.org/10.1016/j.rse.2019.111380">https://doi.org/10.1016/j.rse.2019.111380</a>).
  - We added this citation.
- 5. There are multiple sites in Alaska where in-situ soil moisture is available, and those should be included, such as the site from Delta Junction (NEON site).
  - We added Delta Junction site to the validation.
- 6. A description of the study site is required in the main text.
  - We added the description of the study site to the manuscript (subsection 2.1).
- 7. It is unclear why CORINE land cover is used. The new ESA 10 m land cover provides more sufficient information for this study and spatially has more detail than CORINE.
  - CORINE has a spatial resolution of 100 m, which is closer to the smallest acquired spatial resolution of 250 m. ESA 10m WorldCover data has unnecessarily high spatial resolution for our study.
- 8. L336: The NISAR mission will provide a 200 m soil moisture product as an operational soil moisture product. It is suggested to include proper citations in the manuscript (<a href="https://doi.org/10.1016/j.rse.2023.113667">https://doi.org/10.1016/j.rse.2024.114288</a>).
  - We added these citations to the manuscript.
- 9. L338: NISAR will be launched in April 2025.
  - We corrected this.

**Answers to reviewer #2**

We thank reviewer #2 for their comments on our manuscript. Our answers to the comments are below in blue, after the reviewer's comments.

In this manuscript, the authors presented a machine-learning-based model to downscale SMAP soil moisture data from 9km to 1km (or 250m). However, the method is limited to forested areas only, without considering peatlands or other land types. Similar to RC1's comments, I believe the manuscript demonstrates good potential for a method to downscale soil moisture mapping with good writing, however, major revisions should be made to improve its quality before publication.

**Major comment:**

- 1. Model performance: while the validation statistics provided do suggest an improvement of correlation with the in-situ data, the graphs in Figure 4 clearly suggest a consistent and considerable underestimation of the predicted data at 3 out of 4 sites compared to the in-situ data (~30-60%). At site B and D, the predicted data are even lower than the raw SMAP. Considering that soil moisture directly translates to "volumetric water content" as mentioned in L65, this underestimation is quite concerning as it unintentionally reduced approximately half of soil water volume in the area and should be addressed. Thus, I urge the authors to conduct additional model calibration (if applicable) and provide additional comment on validation results on p-bias as well as reasons why this underestimation happened.
  - Yes, this is a good observation. We replaced the ERA5-Land data with higher resolution (1 km) precipitation and temperature data from <a href="https://en.ilmatieteenlaitos.fi/gridded-observations-on-aws-s3">https://en.ilmatieteenlaitos.fi/gridded-observations-on-aws-s3</a>), which improved the results. We also added a table (Table 3) where the mean relative differences between different test sites and the whole test set are included.
- 2. On similar issue, considering that the in-situ soil moisture data is available starting from 1952 (as mentioned in L133), I believe it will greatly improve the manuscript and method novelty/trustworthy if the authors could generate a longer dataset and validate with available (despite gaps at some stations) in situ data rather than just the summer or 2019.

- ISMN has data in some stations starting from 1952, but unfortunately, the in situ sites in Alaska have much less temporal coverage. The year 2019 was chosen because all the selected sites (forest and others) have data during summer 2019, as we wanted to keep the comparisons comparable. We added more data from other years to validation (L. 150-159; subsection 4.3; Figures 9 and 10, Table A2).
- 3. Further examination of Figure 4 suggests that the raw SMAP has considerable noise compared to in-situ data, which is then transferred to the downscaled product as mentioned by the authors in L295. While I agree that clearing these noises is not a straightforward task, I believe it is not entirely out of scope for this study considering various forcing data such as precipitation and temperature are already included as input for soil moisture modeling. Thus, I suggest the authors provide additional analyses investigating if there are any linkages between the spikes in insitu soil moisture data with the precipitation/temperature dataset from ERA5-Land. Both of these forcing clearly have a direct impact on soil moisture. Thus, if there are some statistical linkages, a modification on how the forcing data is processed as input for the soil moisture is needed and should further improve the novelty of this new method by directly reducing noise of the raw SMAP data.
  - We investigated if there were any linkages between the variation in in situ soil moisture data and higher resolution precipitation and temperature data, but we could not find a clear relationship between precipitation and temperature with soil moisture.
- 4. The abstract mention that the method can provide "high temporal" soil moisture data, however, I fail to find some information or a clear comparison on temporal coverage (time step, start, end, etc.) for any of the referenced dataset (including in situ) or the newly generated data within the introduction/data section. Please consider adding additional information in the introduction section and provide a paragraph in discussion, as well as some mentioning in conclusion section comparing the significant improvement in temporal coverage of this new dataset with raw SMAP and the currently available downscaled ones (Zheng et al., 2023, Zhang et al., 2023, Fang et al., 2022, etc.).
  - Apologies, the term "high temporal" is misleading. It was connected to the possibility of having daily (or even twice a day, if SMAP 6 pm data had been used) soil moisture data in high resolution over boreal forests. We removed the mention of high temporal soil moisture data from the abstract.

**Minor comment:**

- 1. L20, consider removing "for example".
  - We removed this.
- 2. L45, "... are not specifically designed for boreal forests", please provide references for this claim and additional information on how the method being presented are more tailored for boreal forests.
  - We modified the text (L. 49-56)
- 3. L278-294, "Even though SMAP... and agriculture", this section clearly discusses about the based input data, thus, should be moved to the data section.
  - Yes, this is true. We moved this part from the discussion section.

**Answers to editor comments**

We also thank editor for their good comments.

The authors also need to do a proper literature review to compare the results against all the available products at 1 km.

- We added a literature review to the manuscript (L. 306-322; Table 6)

The actual resolution of SMAP 9 km gridded product is  $\sim$ 33 km. Thus, in the study it recommended to use it as  $\sim$ 33 km instead of 9 km. It is possible to obtain fair results for wrong reasons.

- This is true. As we were not sure how to implement the original 33 km resolution information, we decided to change the SMAP at 9 km resolution to the SMAP at 36 km resolution. This led to major rewritings and changes to the model, data sets, and results (e.g. subsection 3.2, section 4)

**List of relevant changes made to the manuscript**

- SMAP data in 9 km spatial resolution was changed to the SMAP data in 36 km spatial resolution
- ERA5-Land data (precipitation, temperature and solar radiation) was removed from the training and test sets
  - Solar radiation was removed altogether from the data sets

- Precipitation and temperature data for training and test sets were replaced with interpolated daily weather observations data at 1 km spatial resolution
- For validation set, similar interpolated data sets were not possible to calculate, and therefore ERA5-Land data was used for temperature, and Global Precipitation Measurement mission data was used for precipitation
- SoilGrids data was removed altogether
- Due to the changes in SMAP data led to changes in in situ data divisions to the training and test sets
- For validation set, one in situ station was added (DEJU from NEON)
  - We also acquired longer time periods of validation data instead just one year
    (2019), and we screened and cleaned the validation data
- Due to the changes in input data, the gradient boosting model was trained again,
  which led to major rewriting and to most of the tables and figures being changed
  - o Rewriting in all sections of the manuscript
  - Changed tables: 1-5 (where Table 3 is new)
  - o Changed figures: 1 (changes in stations); 2-10 (where Figure 9 is new)
- Table 6 was added to help with the literature review of 1 km soil moisture products